# Ricci Curvature, Robustness, and Causal Inference on Networked Data

## Abstract

In the complex landscape of networked data, understanding the causal effects of interventions is a critical challenge with implications across various domains. Graph Neural Networks (GNNs) have emerged as a powerful tool for capturing complex dependencies, yet the potential of geometric deep learning for GNN-based network causal inference remains underexplored. This work makes three key contributions to bridge this gap. First, we establish a theoretical connection between graph curvature and causal inference, revealing that negative curvatures pose challenges in identifying causal effects. Second, based on this theoretical insight, we present computational results using Ricci curvature to predict the reliability of causal effect estimations, empirically demonstrating that positive curvature regions yield more accurate estimations. Lastly, we propose a method using Ricci flow to improve treatment effect estimation on networked data, showing superior performance by reducing error through flattening the edges in the network. Our findings open new avenues for leveraging geometry in causal effect estimation, offering insights and tools that enhance the performance of GNNs in causal inference tasks.

## 1 Introduction

Estimating the causal effect of an intervention on a unit based on observational data is a fundamental task in various domains with far-reaching implications for policy making. This includes fields such as epidemiology, medicine, economics, and political science (Rothman & Greenland, 2005; Yao et al., 2021; Varian, 2016; Keele, 2015). The advent of deep learning has revolutionized numerous disciplines, and its techniques have begun to make inroads into the domain of causal inference (Louizos et al., 2017; Pawlowski et al., 2020; Kallus, 2020; Luo et al., 2020). Causal treatment effect estimation methods aim to estimate causal quantities by statistical ones (Pearl, 2009a). Due to the endogeneity in the network structure, identifying causal effects is particularly challenging on a network of units with non-trivial dependencies (van der Laan, 2012; Zheleva & Arbour, 2021). Designed to learn from networked data, Graph Neural Networks (GNNs) have shown great promise in a variety of applications, including social network analysis, recommendation systems, and biological network analysis (Sperduti, 1993; Frasconi et al., 1998; Gori et al., 2005; Scarselli et al., 2008; Kipf & Welling, 2016; Gilmer et al., 2017; Zhou et al., 2020; Wu et al., 2020; Zecevic et al., 2021). Recently, the success of GNNs in learning from networked data has been extended to network causal inference (Wein et al., 2021); GNN-based causal effect estimation has been proposed to account for network-induced endogeneity in structured observational data (Jiang & Sun, 2022; Kaddour et al., 2021; Cristali & Veitch, 2022; Ma & Tresp, 2021; Guo et al., 2020; Harada & Kashima, 2021).

Despite these advances, the power of GNNs is yet to be unleashed for causal inference on networked data. In the realm of geometric deep learning, GNNs enable the leveraging of inherent geometry in graph-structured data (Bruna et al., 2014; Bronstein et al., 2017; Monti et al., 2017; Cao et al., 2020; Gong et al., 2020; Ye et al., 2020; Bronstein et al., 2021; Atz et al., 2021). For example, discrete curvature on graphs has been used to alleviate issues with over-squashing and over-squeezing in GNNs (Topping et al., 2021), or to devise a distance measure between graphs, with applications in generative GNNs (Southern et al., 2023). However, the relationship between the geometric properties of the graph, such as its curvature, and the performance of the corresponding GNN in estimating causal effects from networked data has not been thoroughly investigated. This omission is significant, as

the geometry of the graph can have a profound impact on the behavior of processes taking place on the graph, and hence on endogeneities rooted in the network.

This work aims to fill this gap by formally establishing the relationship between graph curvature and causal inference on networked observational data. We explore this connection both through theoretical results pointing to such a relationship, and through theoretically-informed experiments illustrating this connection using a GNN-based causal effect estimation method. Drawing from the theory of invariance of causal models and the distributional robustness formulation of causal invariance (Peters et al., 2016; Meinshausen, 2018; Bühlmann, 2020; Weichwald & Peters, 2021), the central premise of this paper is inspired by the proposition that curvature could serve as a practical measure for robustness in networks (Demetrius & Manke, 2005; Tannenbaum et al., 2015). Recent studies have established the connections between curvature, robustness, and entropy (Tannenbaum et al., 2015; Sandhu et al., 2015; Pouryahya et al., 2017). Meanwhile, the relationship between entropy and causal inference has been explored in the context of causal discovery (Compton et al., 2022). Collectively, these works and the foundations here developed suggest that graph curvature could offer a powerful tool for enhancing the performance of GNNs in causal inference tasks.

We present a theoretical layout of causal inference from a distributional robustness perspective, entropic causal inference, and curvature as a robustness indicator, which prepares the ground for establishing the connection between curvature and causal inference. This connection is formally implied from our Theorem 2, which suggests that identification of causal effects becomes more challenging where the curvature is negative. Applying this theoretical finding to causal inference on empirical networks using GNNs, our experiments show that treatment effect estimation error is lower in regions with non-negative curvature, firmly validating our theoretical foundations. Lastly, we propose an adjustment using the Ricci flow to flatten the network, which leads to a remarkable gain in estimating treatment effects on observational networked data.

### Main Contributions

**Theoretical Foundations:** We establish a theoretical connection between curvature and causal inference on networks. Specifically, we show that identifying the causal effect is more challenging in regions of the network with highly negative curvatures. This insight provides a foundational understanding of how the geometric properties of a network can influence causal analysis.

**Experimental Results:** Guided by our theoretical findings, we present computational results that use the Ricci curvature on graphs to predict where causal effect estimates are most reliable. In strong concordance with our theoretical expectations, our empirical results show that the estimation of the treatment effect tends to be most accurate in areas of the network with positive curvature. This demonstrates the practical applicability of our theory in real-world scenarios.

**Methodological Contribution:** We propose a novel method using the Ricci flow for improving the estimation of treatment effect on networked data. This method involves preprocessing the data through a weight adjustment that flattens the network using the Ricci flow. Our proposed method leads to superior performance in estimating treatment effects on observational networked data. This offers a new tool for enhancing the accuracy and reliability of causal inference in complex networks.

## 2 Causality, Invariance, and Robustness

### 2.1 Preliminaries

Consider an outcome of interest $Y$ for a unit with features $X$. We are interested in evaluating the causal effect of a treatment $T$ on $Y$, which can be measured for each individual unit $i$ by the individual treatment effect (ITE), or the expected treatment effect conditioned on the features, known as the conditional average treatment effect (CATE). Given features $x_i$ of an individual, the CATE is given by $\tau_i(x_i) \coloneqq \mathbb{E}\left[Y_i | do(t_i = t) - Y_i | do(t_i = t') | x_i\right]$, where $Y_i | do(t_i)$ is the potential outcome of the unit upon intervention by treatment $t_i$, represented by the $do(.)$ operation (Pearl, 2009b). Following Shalit et al. (2017) and Jiang & Sun (2022), we adopt a conditional formulation of the ITE as the CATE for the features of an individual unit, and throughout our experiments, we refer to $\tau_i(x_i)$ as ITE. Since the data is missing the *counterfactual outcome*, $\tau_i(x_i)$ is only a causal quantity and cannot be directly computed as a statistical quantity. This is referred to as *the fundamental prob-*

*lem of causal inference* (Holland, 1986). Hence, causal effect estimation is essentially to estimate causal quantities from statistical quantities. Whether this estimation is possible —the *identification* problem— is the central question of causal inference (Pearl, 2003).

Identification of the causal effect from the data is contingent on the assumptions we make; common ones are *positivity*, *consistency*, *ignorability*, and *stable unit treatment value assumption (SUTVA)* (Rubin, 1980; Rosenbaum & Rubin, 1983; Imbens & Rubin, 2015; Forastiere et al., 2021; Jiang & Sun, 2022), which are formally defined in Appendix A. When estimating causal quantities on a network of units, relaxing ignorability and SUTVA is likely essential due to peer effects on each unit from its neighbors' features and treatments (Jiang & Sun, 2022).

## 2.2 CAUSAL INFERENCE AS RISK MINIMIZATION

Following Bühlmann (2020), we formalize the derivation of causal quantities from statistical quantities as a worst-case risk minimization problem. We combine treatment and covariates (features) in one variable, denoted $\mathbf{X}$, and the outcome is denoted by $\mathbf{Y}$. Adopting the notation in Bühlmann (2020), let $Y^e$ and $X^e$ denote the random variable and random vector corresponding to an observed environment $e \in \mathcal{E}$, and let $\mathcal{F} \supseteq \mathcal{E}$ denote the union of observed and unobserved environments encompassing the joint distribution of $\mathbf{X}$ and $\mathbf{Y}$. The causal relationship of $\mathbf{X}$ and $\mathbf{Y}$ is trivially revealed when $\mathcal{F} = \mathcal{E}$, hence, without loss of generality, we assume $\mathcal{E} \subset \mathcal{F}$.

Learning the relationship between $\mathbf{X}$ and $\mathbf{Y}$ can be described as predicting $Y^e$ from $X^e$ based on observations $e \in \mathcal{E}$, such that the prediction is robust under the choice of $e \in \mathcal{F}$. To this end, consider a linear model as an example; we can formulate a causal inference parameter, $\theta_{\text{causal}}$, as the worst case regression estimand below, with the constraint that $e$ does not directly impact the joint distribution of $X^e$ and $Y^e$ (Bühlmann, 2020), hereafter referred to by $\mathbf{C}$,

$$\theta_{\text{causal}} = \arg \min_b \max_{e \in \mathcal{F}} \mathbb{E}\left[\left(Y^e - (X^e)^T b\right)^2\right]. \tag{1}$$

## 2.3 INVARIANCE OF CAUSAL MODELS

Invariance of this worst-case risk minimization is a core component behind inferring causality from data. Given a set of environments $\mathcal{G} \subseteq \mathcal{F}$, invariance can be formalized as the existence of a subset of covariate indices $S \subset \{1, \ldots, n_X\}$ satisfying $\mathbf{A}_S(\mathcal{G})$, defined below,

**Definition 1** (Bühlmann, 2020). $\mathbf{A}_S(\mathcal{G})$ *is defined as the property that* $\{\mathcal{L}(Y^e | X_S^e) | e \in \mathcal{G}\}$ *is a singleton, where* $X_S^e$ *denotes the subset of covariates induced by indices in* $S$, *and* $\mathcal{L}(Y^e | X_S^e)$ *denotes the loss function* $\mathbb{E}\left[\left(Y^e - (X_S^e)^T b\right)^2\right]$.

If $\mathbf{A}_S(\mathcal{G})$ holds, the causal parameter in Equation 1 remains the same under variations in $e \in \mathcal{G}$. For causal inference, we are particularly interested in the invariance assumption $\mathbf{A}_S(\mathcal{G})$ when $\mathcal{G} = \mathcal{E}$ for estimating $\theta_{\text{causal}}$ from the data, or when $\mathcal{G} = \mathcal{F}$ for the more general case of determining causal parameters over the population. Assuming there exists an $S$ for which $\mathbf{A}_S(\mathcal{F})$ holds, the problem of causal inference is then to find such $S = \text{pa}(Y) \subset \{1, \ldots, n_X\}$, where $\{X_i\}_{i \in \text{pa}(Y)}$ is the set of direct causal parents of $Y$. Taking a step towards computation, this problem can be formulated in terms of structural equation models (SEMs) between $X$ and $Y$, as finding the set $\text{pa}(Y)$ such that $\mathbf{C}$ is satisfied (Bühlmann, 2020). This can be formalized as satisfying $\mathbf{B}(\mathcal{F})$, where $\mathbf{B}(\mathcal{G})$ is,

**Definition 2** (Bühlmann, 2020). $\mathbf{B}(\mathcal{G})$ *is defined as the property that* $\left\{p_{\epsilon^e} | e \in \mathcal{G} \wedge Y^e = f\left(X_{pa(Y)}^e, \epsilon^e\right)\right\}$ *is a singleton, where* $f$ *determines the SEM,* $\epsilon^e$ *is independent of* $X_{pa(Y)}^e$, *and* $p_{\epsilon^e}$ *is the distribution of* $\epsilon^e$.

The assumption $\mathbf{B}(\mathcal{F})$ in fact completes the formulation of causal inference problems from the perspective of invariance, with Proposition 1 in Bühlmann (2020), which states that under $\mathbf{B}(\mathcal{F})$, $\text{pa}(Y)$ satisfies $\mathbf{A}_{\text{pa}(Y)}(\mathcal{F})$. It follows that an identification strategy, when computing $\theta_{\text{causal}}$ over the observed environments, is taking the intersection of all sets $S$ satisfying $\mathbf{A}_S(\mathcal{E})$. The main issue, however, is that such an identification mechanism relies on assumption $\mathbf{B}(\mathcal{F})$ and condition $\mathbf{C}$. We next discuss the robustness of an estimator in a regression problem which allows for relaxing these constraints.

## 2.4 DISTRIBUTIONAL ROBUSTNESS AND CAUSAL INFERENCE

One common situation where **C** fails is the presence of hidden confounders. We can use anchor regression (Rothenhäusler et al., 2021) to relax **C** and allow for hidden confounders. In anchor regression, we consider an anchor variable $A$ with $pa(A) = \emptyset$. We allow $A$ to be a causal parent of the covariates $X$, outcome $Y$, and hidden confounders $H$, as described in Figure 1. The anchor variable could be considered as an environment that is not constrained by **C**. The corresponding linear SEM is then

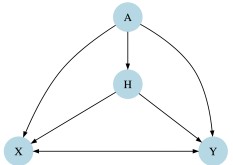

$$\begin{bmatrix} X \\ Y \\ H \end{bmatrix} = B \begin{bmatrix} X \\ Y \\ H \end{bmatrix} + \epsilon + MA, \tag{2}$$

where $B$ and $M$ are unknown constant real matrices and $\epsilon$ is the noise vector which satisfies $\epsilon \perp\!\!\!\perp A$. This yields the following anchor regression problem for regressing $Y$ on $X$,

Figure 1: Causal graph for the anchor regression model. $A$, $H$, $X$, and $Y$ denote the anchor, hidden confounders, covariates, and outcome.

$$Y = X^T\beta + H^T\alpha + A^T\xi + \epsilon_Y. \tag{3}$$

Since the anchor variable is a source in the graphical model, the anchor regression estimator minimizes a risk in the column space of $A$. Let $\Pi_A$ be the projection matrix onto the column space of $A$ for the sample, and let $P_A$ denote the corresponding projection operator for the population case. The anchor regression estimand $\beta_A(\gamma)$ and estimator $\hat{\beta}_A(\gamma)$ for regressing an $n \times 1$ outcome $\mathbf{Y}$ on an $n \times m$ matrix of covariates $\mathbf{X}$, corresponding to $Y$ and $X$ in Equation 3, are given by

$$\beta_A(\gamma) = \arg\min_b \left\{ \mathbb{E}\left[\left((I - P_A)(Y - X^Tb)\right)^2\right] + \gamma\mathbb{E}\left[\left(P_A(Y - X^Tb)\right)^2\right]\right\}, \tag{4}$$

$$\hat{\beta}_A(\gamma) = \arg\min_b \left\{ \frac{1}{n}\left\|(I - \Pi_A)(\mathbf{Y} - \mathbf{X}b)\right\|_2^2 + \frac{\gamma}{n}\left\|\Pi_A(\mathbf{Y} - \mathbf{X}b)\right\|_2^2\right\}, \tag{5}$$

where the second term in the objective functions encourages the residuals to be orthogonal to $A$ (Bühlmann, 2020). We can compute $\hat{\beta}_A(\gamma)$ through the Ordinary Least Square estimator for regressing a transformed outcome variable $W_\gamma Y$ on the corresponding transformed covariate $W_\gamma X$, where $W_\gamma := I - \left(1 - \sqrt{\gamma}\right)\Pi_A$. Recall that $A$ captures the influence of what we previously referred to as the environment, thus encouraging independence of residuals from the environment and leading to further invariance with respect to the environment.

Consider the system under perturbation by a vector $v = M\delta$ for some $\delta$ replacing the anchor term in Equation 2. The SEM under perturbation can be written as

$$\begin{bmatrix} X^v \\ Y^v \\ H^v \end{bmatrix} = B \begin{bmatrix} X^v \\ Y^v \\ H^v \end{bmatrix} + \epsilon + v. \tag{6}$$

Let us impose $\delta \perp\!\!\!\perp \epsilon$ and constrain the norm of the expected perturbation by the order of a constant $\gamma$. That is, we consider a class of shift perturbations $\mathcal{C}_\gamma$ where the perturbation is generated in the column space of $M$ by a vector $\delta$ independent of the noise, and where the typical size of the perturbation is $O(\gamma)$ as $\gamma \to \infty$. Also assume, without loss of generality, that $X$ and $Y$ are centered at 0. Under these conditions, if $\mathbb{E}\left[AA^T\right]$ is positive definite, the following proposition holds (Rothenhäusler et al., 2021; Bühlmann, 2020).

**Proposition 1.** *Given any $b \in \mathbb{R}^m$, if $A$ and $Y - X^Tb$ are uncorrelated, $Y^v - (X^v)^Tb$ in the perturbed system has the same distribution for all $v \in span(M)$.*

Proposition 1 points to what leads to the distributional robustness of the anchor regression estimand. This is due to an equality between a worst-case residual in the perturbed system and the objective function for the estimand in Equation 4 (Rothenhäusler et al., 2021; Bühlmann, 2020),

**Theorem 1.** *For any $b \in \mathbb{R}^m$*

$$\sup_{v \in \mathcal{C}_\gamma} \mathbb{E}\left[\left(Y^v - (X^v)^Tb\right)^2\right] = \mathbb{E}\left[\left((I - P_A)(Y - X^Tb)\right)^2\right] + \gamma\mathbb{E}\left[\left(P_A(Y - X^Tb)\right)^2\right].$$

Corollary 1, which states that $\beta_A(\gamma)$ minimizes a worst case risk over the class of shift perturbations $\mathcal{C}_\gamma$, follows trivially considering Equation 4:

**Corollary 1.** $\beta_A(\gamma) = \arg\min_{b \in \mathbb{R}^m} \sup_{v \in \mathcal{C}_\gamma} \mathbb{E}\left[\left(Y^v - (X^v)^T b\right)^2\right].$

Recall that the second term in the objective function of the anchor regression estimand in Equation 4 is essentially a causal regularization term that encourages the invariance of the residuals with respect to the environment. Theorem 1 and Corollary 1 establish that the anchor regression estimand corresponds to a worst-case risk minimization in a perturbed system, and simultaneously encourages conditions which bring us closer to a scenario where the assumptions for causal identification hold. In other words, an estimator that satisfies the criteria for a causal parameter is also a distributionally robust optimizer. This concludes our discussion of causal inference as a worst-case risk optimization, establishing the connection between causal inference and distributional robustness.

## 3 CURVATURE, ROBUSTNESS, AND ENTROPY

We now detail why we expect the Ricci curvature to be related to causal inference. Curvature controls how volume balls and geodesics on a Riemannian manifold behave in their local neighborhood (Do Carmo & Flaherty Francis, 1992). The Ricci curvature indicates how much the local geometry induced by a Riemannian metric deviates from that of a Euclidean space (Pouryahya et al., 2017). In particular, the lower bounds of the Ricci curvature provide an estimate on the tendency of the volumes to differ locally from the Euclidean volume (Bauer et al., 2017). Extended to discrete structures such as graphs, the graph Ricci curvature characterizes deviation of the neighborhood of an edge from a grid, capturing the dispersion through the edge in its neighborhood. Ricci curvatures on graphs have been proven powerful for performing various computational tasks on graph neural networks (Topping et al., 2021; Southern et al., 2023; Di Giovanni et al., 2023; Liu et al., 2023). For the experiments in this paper, we use the Ollivier-Ricci curvature (Ollivier, 2009) —a graph curvature notion rooted in optimal transport. A formal definition of Ollivier-Ricci curvature as well as an alternative Ricc-type curvature are included in Appendix B.

### 3.1 CURVATURE AND ENTROPY

The following result from optimal transport Lott & Villani (2009), offers bounds on the Boltzmann entropy in terms of a lower bound on the Ricci curvature,

$$S(\mu_\lambda) \geq (1 - \lambda) S(\mu_0) + \lambda S(\mu_1) + \underline{k} \frac{\lambda(1 - \lambda)}{2} W_2(\mu_0, \mu_1)^2,\tag{7}$$

where $S(.)$ denotes the Boltzmann entropy (Adkins, 1983), $\underline{k}$ is a lower bound on the Ricci curvature, $W_2(\mu_0, \mu_1)$ is the Wasserstein distance of order 2 between $\mu_0$ and $\mu_1$ in the metric space $(P(\mathcal{X}), W_2)$ of probability measures on $\mathcal{X}$, and $\mu_\lambda$ for $\lambda \in [0, 1]$ gives the geodesic between them (Pouryahya et al., 2017). This inequality indicates a positive correlation between Ricci curvature $k_R$ and entropy (Pouryahya et al., 2017), i.e.,

$$\Delta S \times \Delta k_R \geq 0.\tag{8}$$

### 3.2 CURVATURE AND ROBUSTNESS

There is a correlation between *system robustness* and entropy, through the fluctuation theorem (Pouryahya et al., 2017). Characterized by the fluctuation decay rate (Demetrius & Manke, 2005), system robustness refers to the ability of the system to rapidly return to its stationary state after a perturbation. Therefore, by the Fluctuation Theorem (Evans et al., 1993), there is a positive correlation between system robustness and entropy, which in turn implies that system robustness is positively correlated with curvature (Pouryahya et al., 2017), considering Equation 8. This points to a connection between Ricci curvature and causal inference, in light of the discussion on distributional robustness and causal inference in Section 2. However, claiming such a connection is premature at this stage, due to the fact that distributional robustness and system robustness are two fundamentally different notions of robustness. Thus, we utilize the correlation between Boltzmann entropy and Ricci curvature to formally establish this anticipated connection. To do so, we use the results from entropic causal inference (Compton et al., 2020), which we briefly review in the following section.

## 4 ENTROPIC CAUSAL INFERENCE

Entropic causal inference is a framework that aims to learn the causal graph between variables from observational data, using an Occam's razor-type principle Kocaoglu et al. (2017). This approach seeks the information-theoretically simplest structural explanation of the data to infer causality (Compton et al., 2020). The central claim is that the true causal structural model is one that yields the minimum entropy (Compton et al., 2022). Under a set of assumptions, this principle is shown to facilitate correct orientation of the edges in the causal graph in a two-variable setting (Compton et al., 2020); and is addressed in the more general case of multi-variable causal graphs in Compton et al. (2022), by finding the minimum entropy coupling between each pair of connected variables.

These results further point to the relationship between Shannon entropy and distributional robustness. Fitting the wrong model to the data requires a higher entropy than the correct model (Compton et al., 2020; 2022). More precisely, let $Y = f(X, E)$ be the structural causal model, where $E \perp\!\!\!\perp X$ denotes exogenous variables. If the entropy $H(E)$ is sufficiently small, for the data to fit an alternative structural model $X = g(Y, \tilde{E})$, with high probability, the Shannon entropy of the alternative exogenous variables $\tilde{E} \perp\!\!\!\perp Y$ is bounded from below,

$$H(X) + H(E) - H(Y) < H(\tilde{E}). \tag{9}$$

Considering the correlation between curvature and entropy, this further points to a connection between curvature and causal inference. We present this connection next.

## 5 CURVATURE AND CAUSAL INFERENCE

The results discussed in Section 3.1 indicate a positive correlation between Boltzmann entropy and Ricci curvature, and in Section 4 we stated a bound on the Shannon entropy of the exogenous variables in an alternative structural causal model, different from the true model. We now show a connection between Ricci curvature and causal inference. This connection will, in turn, inform a methodological remedy utilizing the Ricci flow to improve treatment effect estimates.

Consider the problem of identifying the causal relationship between $X_i$ and $Y_i$ for $i \in \{1, 2\}$, corresponding to two sets of data with the true causal models given by $Y_i = f_i(X_i, E_i)$. Suppose an alternative model $X_i = g(Y_i, \tilde{E}_i)$, with alternative exogenous variables $\tilde{E}_i$ fits the data, and assume that the conditions described in Section 4 leading to Equation 9 are satisfied. Assume that the Ricci curvature corresponding to $X_i$ is bounded below by $\underline{k}_i$, for $i \in \{1, 2\}$. Then, under the assumptions stated in Appendix C, where we provide the proof, the following holds:

**Theorem 2.** *If $\underline{k}_1 < 0 \leq \underline{k}_2$, there exists a value $\eta$, for which $\mathbb{P}\left[H(\tilde{E}_2) > \eta\right] \geq \mathbb{P}\left[H(\tilde{E}_1) > \eta\right]$, i.e., the probability that the Shannon entropy of $\tilde{E}_2$ is lower bounded by $\eta$ is at least as high as the probability that $\eta$ is a lower bound for the Shannon entropy of $\tilde{E}_1$.*

Theorem 2 states that if the lower bound on the Ricci curvature is negative for $X_1$ and non-negative for $X_2$, then the alternative exogenous variables in the wrong causal model are more likely to need a larger entropy when fitting $(X_2, Y_2)$ than $(X_1, Y_1)$. In other words, this theorem implies that for the wrong model to fit the data, we expect a higher entropy of the exogenous variables when the curvature is larger, and in particular non-negative, as opposed to negative.

Theorem 2 helps us establish the connection between Ricci curvature and causal inference, from the perspective of distributional robustness. Specifically, the theorem implies that when the curvature is positive, a smaller class of exogenous variables could make the wrong model fit the observations. This means that a positive curvature leads to higher distributional robustness, making the worst-case risk minimization in a system under perturbation a less challenging problem. In other words, when the Ricci curvature corresponding to the covariates is positive, the regression estimator is identified for a larger class of perturbations. This ultimately suggests that more positive Ricci curvatures are expected to correspond to lower errors in estimating the causal effect, concluding the main goal of our theory, which establishes for the first time a formal connection between the geometry of networks and causal inference. We next show how to use this foundational result to improve causal estimation, and in Section 7, we illustrate this with experiments.

## 5.1 Ricci Flow Adjustment for Improving Causal Effect Estimates

Informed by the theoretical connection between Ricci curvature and causal inference, we propose to improve treatment effect estimates on network data using the discrete Ricci flow (Jin et al., 2008; Ni et al., 2019). Given a Riemannian metric $g$, under the Ricci flow, at time $t$, $g$ evolves as $\frac{\partial g_{ij}(t)}{\partial t} = -2R_{ij}$, where $R_{ij}$ is the Ricci curvature. This is, in fact, equivalent to the heat equation, considering the formulation of the Ricci flow as a scaling of the Laplacian of the metric tensor (Chow & Knopf, 2004). As a result, just as the temperature evolves towards a more uniform distribution under heat diffusion, the Ricci flow evolves to a uniform distribution of curvature (Jin et al., 2008; Hamilton, 1988). Similarly, the discrete analog of the Ricci flow on a graph leads to a flatter network, with the majority of edges having near-zero Ricci curvature. This discrete analogue of the flow is defined as

$$w_{vu}^{i+1} = (1 - \kappa_{vu}) \, d(v, u)^i, \tag{10}$$

where $w_{vu}$ is the weight on the edge $(v, u) \in E$, $\kappa_{vu}$ is the discrete Ricci curvature, $d(v, u)$ is the geodesic distance, and $i$ is the iteration index.

In order to improve estimations of treatment effects, we propose modifying the edge weights via the discrete Ricci flow to obtain an adjusted shift operator for the graph convolution, which is the weighted adjacency matrix. This is, in essence, preprocessing the training data through computing a weight matrix by which we multiply the adjacency matrix, and hence, a cost-efficient one-time computation. Since real-world networks are predominantly sparse, this flattening increases the Ricci curvature of the majority of the edges in the network and, therefore, based on our theory, is expected to reduce the error in estimating causal treatment effects.

## 6 Related Work

We showed the connection between curvature and causal inference, bridging for the first time the works on invariance and robustness of causal models, geometric deep learning, and deep learning for causal inference. While the related work has been cited in each section as needed, in Appendix D we mention the most relevant works in each aspect and point to other works in the literature. Although these works provide critical foundations and motivations for the theory in this work, none of them makes the explicit connections we developed in this paper and, in particular, the close connection between geometry/curvature, robustness, and causal inference.

## 7 Experiments

Building upon these theoretical foundations, we now turn our attention to empirical validation. We employ numerical experiments on real-world data to demonstrate the practical utility of Ricci curvature for causal effect estimation on networked data. Considering the success of neural networks in estimating causal treatment effect in nonlinear systems, we use a GNN-based framework to obtain causal effect estimates of the treatment on the nodes in networked data.

## 7.1 Model and Data

A causal model determines a relationship between treatments, features, and outcomes, while a statistical model that estimates such a causal model uses the observed outcomes. When treatments are applied to a network with non-trivial connections, traditional causal effect estimation methods fail due to violation of ignorability or STUVA (Kaddour et al., 2021; Jiang & Sun, 2022; Chu et al., 2023). Jiang & Sun (2022) proposed *NetEst*, a GNN-based model we use here, which yields identifiable estimates of the treatment effect on networked data in settings where SUTVA is violated due to peer exposure effect. Details of the NetEst model, the ITE formulation, the training loss, and implementation are included in Appendix E and Appendix F. Our experiments are primarily aimed at demonstrating our theoretical results in practice, and evaluating the performance of NetEst and our proposed enhancement of it. Additionally, in Section 7.3 we compare our results with several baseline causal effect estimation methods. These baselines include CFR (Shalit et al., 2017), TARNet (Shalit et al., 2017), NetDeconf (Guo et al., 2020), T-Learner and X-Learner Künzel et al. (2019) with random forest (RF) regressors, and T-Learner and X-Learner implemented using a GNN encoder followed by a multilayer perceptron. To evaluate the performance in estimating the treatment

effects, we use the ITE error $\varepsilon_{ITE}(v) := |\tau_v - \hat{\tau}_v|$ and the Precision in Estimation of Heterogeneous Effect (PEHE) $\epsilon_{PEHE} := \sqrt{\frac{1}{N}\sum_{v \in V}(\tau_v - \hat{\tau}_v)^2}$, where $\tau_v$ and $\hat{\tau}_v$ denote the true and estimated ITEs for node $v$.

Consistent with standard practice in causal representation learning, we use semi-synthetic datasets, namely an empirically observed network structure and features with simulated treatments and potential outcomes (Hill, 2011; Shalit et al., 2017; Veitch et al., 2019; Guo et al., 2020; Ma et al., 2021; Jiang & Sun, 2022). Following the original experiments on NetEst, we use the BlogCatalog (BC) and Flickr datasets (Guo et al., 2020; Ma et al., 2021). We supplement our experiments with numerous other empirical networks. All datasets are described in Appendix G.

## 7.2 RICCI CURVATURE AND TREATMENT EFFECT ESTIMATION ERROR

We demonstrate the implications of Theorem 2 by inspecting the joint distribution of $\varepsilon_{ITE}$ and the Ricci curvature. Ollivier-Ricci curvature is inherently an edge-based measure (see Appendix B). To quantify the curvature of the region surrounding a node, we aggregate the curvature of its incident edges by taking their sum. We then compute the empirical joint distribution of the sum of edge curvatures and $\varepsilon_{ITE}$ for each node. The joint distributions in Figure 2 show a negative correlation between Ricci curvature and $\varepsilon_{ITE}$, indicating that treatment effect estimations are more reliable in regions with non-negative curvature. Additional experiments in Appendix H show that these results are consistent not only across different datasets but also different notions of Ricci curvature, confirming our theoretical results.

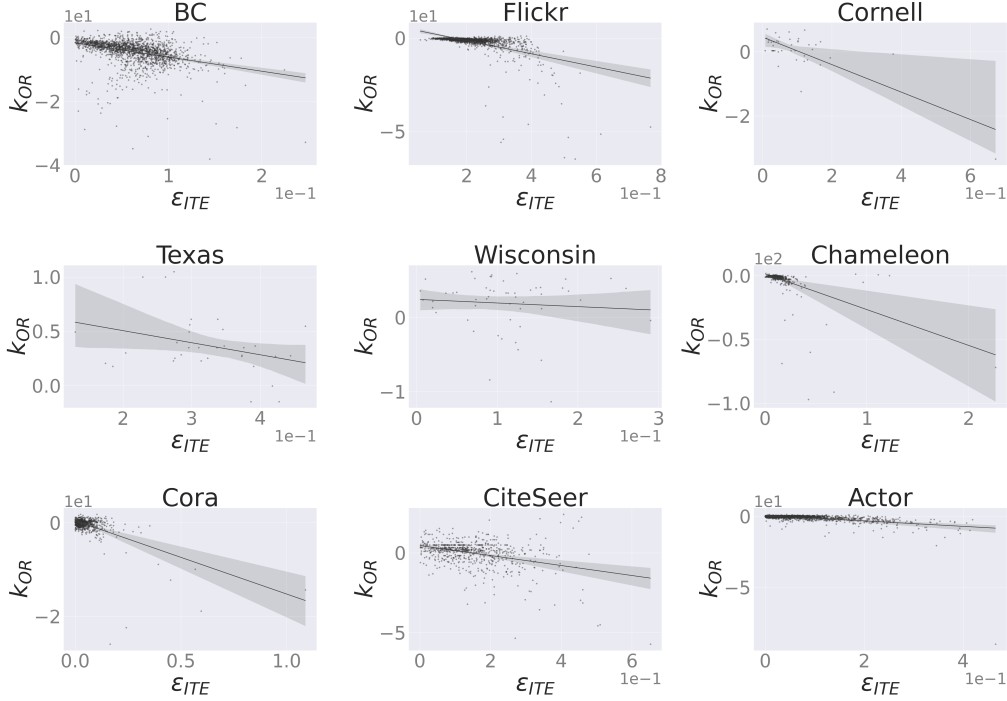

Figure 2: Joint distributions of the sum of Ollivier-Ricci curvatures in the neighborhood of each node and the estimation error of ITE for that node. The distributions for nine different networks are shown (all datasets are described in Appendix G). The regression lines with the corresponding 95% confidence intervals are marked on the plots.

## 7.3 RICCI FLOW ADJUSTMENT FOR TREATMENT EFFECT ESTIMATION

The theory and experiments alike speak to the adverse effect of highly negative curvatures on estimating treatment effects. In line with this observation, in Section 5.1 we proposed a simple method to improve the estimation of treatment effects on networked data by flattening the network via the

discrete Ricci flow. To evaluate this method, we apply this adjustment to the input graph of NetEst. We refer to the modified method as *f-NetEst*. The $\epsilon_{PEHE}$ values obtained from our experiments, Table 1, show that f-NetEst achieves the best performance on all datasets with relative gains of up to 52%. Comparing the distributions of the ITE estimation errors (Appendix H.2) further confirms that the Ricci flow adjustment leads to more accurate ITE estimations. Table 1 also reports the performances of several baseline models. While our experiments primarily focus on NetEst, which outperforms all the baselines, we also explore the impact of the Ricci flow adjustment on the performance of three baseline models featuring GNN encoders: T-Learner+GNN, X-Learner+GNN, and NetDeconf. These additional experiments confirm that our proposed modification results in a reduction in the treatment effect estimation error in most cases across other GNN-based models as well.

Table 1: $\epsilon_{PEHE}$ for nine datasets, comparing the proposed f-NetEst against NetEst and baseline models. The baselines include three models implemented with GNN encoders. The experiment with the Ricci flow adjustment for these models is marked with "f-". Boldface and underline mark the best and second best performances. Green and yellow mark relative gains greater than 5% and less than −5% from the Ricci flow adjustment.

|  | BC | Flickr | Cornell | Texas | Wisconsin | chameleon | Cora | CiteSeer | Actor |
|---|---|---|---|---|---|---|---|---|---|
| T-Learner+RF | 0.328 | 0.462 | 0.192 | 0.414 | 0.463 | 0.372 | 0.232 | 0.386 | 0.238 |
| X-Learner+RF | 5.612 | 5.745 | 5.928 | 3.827 | 3.815 | 3.709 | 8.626 | 5.606 | 5.231 |
| TARNet | 0.969 | 1.024 | 0.705 | 1.028 | 0.711 | 1.212 | 0.679 | 0.638 | 0.796 |
| CFR | 0.895 | 0.960 | 0.806 | 1.038 | 0.849 | 0.926 | 0.570 | 0.620 | 0.735 |
| T-Learner+GNN | 4.178 | 9.630 | 5.125 | 4.437 | 0.559 | 16.715 | 0.285 | 0.529 | 7.912 |
| X-Learner+GNN | 4.627 | 3.933 | 20.461 | 1.995 | 16.244 | 329.959 | 3.165 | 4.428 | 4.296 |
| NetDeconf | 1.092 | 1.251 | 0.900 | 1.137 | 0.952 | 1.207 | 0.791 | 0.752 | 0.895 |
| f-TLearner+GNN | 3.268 | 2.762 | 4.370 | 3.106 | 0.466 | 7.764 | 0.263 | 0.494 | 3.896 |
| f-XLearner+GNN | 4.222 | 3.859 | 17.395 | 2.020 | 20.815 | 251.290 | 3.053 | 3.919 | 3.967 |
| f-NetDeconf | 1.088 | 1.245 | 0.900 | 1.143 | 0.954 | 1.200 | 0.810 | 0.767 | 0.898 |
| NetEst | 0.069 | 0.213 | 0.165 | 0.330 | 0.147 | 0.247 | 0.082 | 0.176 | 0.094 |
| f-NetEst | **0.033** | **0.208** | **0.127** | **0.308** | **0.142** | **0.230** | **0.078** | **0.165** | **0.088** |

# 8 CONCLUSIONS, LIMITATIONS, AND ETHICAL CONSIDERATIONS

We delved into the unexplored territory of leveraging geometry for causal inference on networked data via GNNs. We established a theoretical connection between curvature and causal inference, uncovering the challenges posed by negative curvatures in identifying causal effects. We presented numerical results using graph Ricci curvature to predict the reliability of causal effect estimations on networked data, empirically validating that positive curvature regions lead to more accurate results. We then proposed using the Ricci flow to enhance treatment effect estimation on networked data, achieving superior performance through flattening the edges in the network. To the best of our knowledge, this work is the first to formally establish the connection between graph curvature and network causal inference; opening new avenues for applications of graph geometry in causal inference, as well as neighboring tasks such as transfer learning, out-of-distribution generalization, and domain adaptation.

**Limitations and future directions.** Our proposed method cannot target specific neighborhoods of the network for improving causal effect estimation. Moreover, using Ricci flow to reduce treatment effect estimation error is a static adjustment on the graph that is not efficiently updated during training. Our proposed improvement effectively alters the graph by weighting the edges, requiring careful consideration regarding conceptual consistency of the edge weights with the context of the problem in hand. In future work, our aim is to incorporate these additional dimensions, enhancing the robustness and applicability of curvature-based techniques in causal inference.

**Ethics statement.** While all used data are standard in the community, they have the risk of being biased, this affecting the experimental results but not the theoretical work. Properly detecting causal factors and their uncertainty, as here introduced, can help with the development of fair ML.

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

## A    CAUSAL IDENTIFICATION ASSUMPTIONS

A set of assumptions, often referred to as *identification strategy*, are commonly considered for identifying the causal effect. In Section 2.1 we named four common assumptions. Here we include a description of these assumptions for completeness (Rubin, 1980; Rosenbaum & Rubin, 1983; Imbens & Rubin, 2015; Forastiere et al., 2021):

- **Positivity:** For every unit $i$, $\mathbb{P}\left[t_i = 1|x_i\right] \in (0, 1)$, i.e., each unit may or may not receive the treatment.

- **Consistency:** If the treatment and covariates of unit $i$ are $t_i$ and $x_i$, then $Y_i = Y_i|do(t_i, x_i)$. In other words, the potential outcome of the observed treatment and covariates is the same as the observed outcome.

- **Strong Ignorability:** Also referred to as *unconfoundedness*, this assumption is formally defined as $\{Y|do(T = 1), Y|do(T = 0)\} \perp\!\!\!\perp T|X$. In other words, conditional on all the measured covariates, the potential outcome does not depend on the treatment assignment.

- **Stable Unit Treatment Values Assumption (SUTVA):** The potential outcome of a unit is unaffected by treatment assignment of all other units.

These assumptions, although not always sufficient or necessary, could lead to identification of the treatment effect in various settings where there is no network effect, but fail to do so in the presence of network effect (Jiang & Sun, 2022). However, Jiang & Sun (2022) show the identifiability of the treatment effect estimated by NetEst, under a set of modified assumptions that account for the covariates of neighbors and the peer effect. For a graph $G = (V, E)$ with treatments $\{t_v\}_{v \in V}$, features $\{x_v\}_{v \in V}$, peer exposures $\{z_v\}_{v \in V}$, and potential outcomes $\{Y_v\}_{v \in V}$, these assumptions are as follows (Jiang & Sun, 2022):

- **Positivity:** For every node $v \in V$, $\mathbb{P}\left[t_v = 1|x_v, \{x_u\}_{u \in N_v}\right] \in (0, 1)$.

- **Consistency:** For every node $v \in V$, $Y_v = Y_v|do(t_v = t, z_v = z)$.

- **Strong Ignorability:** For every node $v \in V$, $Y_v|do(t_v, z_v) \perp\!\!\!\perp t_v, z_v|x_v, \{x_u\}_{u \in N_v}$.

- **Markov:** For any two sets of treatments $\{t_v\}_{v \in V}$ and $\{t'_v\}_{v \in V}$, given any node $w \in V$, if $t_w = t'_w$ and $Z(\{t_u\}_{u \in N_w}) = Z(\{t'_u\}_{u \in N_w})$, then $Y_w|do(\{t_v\}_{v \in V}) = Y_w|do(\{t'_v\}_{v \in V})$, where $Z(.)$ is the exposure function, and we use $do(\{t_v\}_{v \in V})$ to denote enforcing all treatments in $\{t_v\}_{v \in V}$. That is, the potential outcome of any node is only affected by its treatment and the treatments of its immediate neighbors.

## B    RICCI CURVATURE NOTIONS ON GRAPHS

The Ricci curvature indicates deviation from the Euclidean space (Do Carmo & Flaherty Francis, 1992; Pouryahya et al., 2017; Bauer et al., 2017). On graphs, this translates to measuring how much the neighborhood of an edge differs from a grid. We used the Ollivier-Ricci curvature (Ollivier, 2009) for the experiments reported in the main text of the paper. Forman-Ricci curvature (Forman, 2003) is an alternative notion of Ricci curvature on graphs, which we use, in addition to Ollivier-Ricci curvature, in supplementary experiments included in Appendix H.1. In this section, we formally define these two Ricci-type graph curvatures.

The Ollivier-Ricci curvature is an optimal transport formulation of the Ricci curvature on graphs. Given a graph $G = (V, E)$, for an edge $(v, u) \in E$, with $\mu_v$ and $\mu_u$ probability measure on the nodes anchoring $(v, u)$, the Ollivier-Ricci curvature is defined as

$$\kappa_{OR}(v, u) := 1 - \frac{W_1(\mu_v, \mu_u)}{d_G(v, u)}, \tag{11}$$

where $d_G(.)$ is a distance metric on $V$ and $W_1$ denotes the 1-Wasserstein distance (Lin et al., 2011; Jost & Liu, 2014). Given the flexibility with respect to the choice of $\mu_v$ and $\mu_u$, the Ollivier-Ricci curvature is a versatile tool for capturing the local geometry of edges in a graph.

The Forman-Ricci curvature is a combinatorial curvature notion. The Forman-Ricci curvature of an edge $(v, u) \in E$ in an undirected graph is given by

$$\kappa_{FR}(v, u) := w_{vu} \left[ \frac{w_v}{w_{vu}} + \frac{w_u}{w_{vu}} - \sum_{(v', u') \in N_v \times N_u} \left( \frac{w_v}{\sqrt{w_{vu} w_{vv'}}} + \frac{w_u}{\sqrt{w_{vu} w_{uu'}}} \right) \right], \qquad (12)$$

where $w_v$ is the weight of the node $v$, $w_{vu}$ is the weight of the edge $(v, u)$, and $N_v$ ist the set of neighbors of the node $v$ (Sreejith et al., 2016; Weber et al., 2017). By convention, all weights are set to 1 in an unweighted graph, in which case the Forman curvature becomes $\kappa_{FR}(v, u) = 4 - d_v - d_u$, where $d_v$ denotes the node degree.

## C  THEOREM DETAILS

In this appendix we provide the proof of Theorem 2, which states the following under the assumptions listed in the appendix Section C.1: Given $X_i$ and $Y_i$ for $i \in \{1, 2\}$, corresponding to two sets of data with causal models $Y_i = f_i(X_i, E_i)$, if an alternative model $X_i = g(Y_i, \tilde{E}_i)$ fits the data, having non-negative and negative lower bounds on the Ricci curvatures corresponding to $X_2$ and $X_1$ implies that for some constant $\eta$, the probability that the Shannon entropy of $\tilde{E}_2$ is greater than $\eta$ is greater than or equal to the probability that the entropy of $\tilde{E}_1$ is lower bounded by $\eta$.

### C.1  ASSUMPTIONS

Given the triplets $(X_1, Y_1, E_1)$ and $(X_2, Y_2, E_2)$, with structural causal models $Y_i = f_i(X_i, E_i)$ for $i = 1, 2$, we make the following assumptions:

(Ai) Considering probability measures $\mu_{X_1}$ and $\mu_{X_2}$ corresponding to $X_1$ and $X_2$, there exists a pair of measures $\mu_0$ and $\mu_1$ such that $\mu_{X_1}$ and $\mu_{X_2}$ are on the geodesics between $\mu_0$ and $\mu_1$ in a 2-Wasserstein metric space.

(Aii) $H(Y_1) \approx H(Y_2)$ and $H(E_1) \approx H(E_2)$, where we use $\approx$ to denote sufficiently close, and $H(.)$ denotes the Shannon entropy.

(Aiii) The conditions for Conjecture 1 in Kocaoglu et al. (2017) and Compton et al. (2020): $X \sim p(X)$ and $E \sim p(E)$, where $p(X)$ is a uniform random sample from the $n$-dimensional probability simplex, $p(E)$ is sampled uniformly from the points in the $m$-dimensional probability simplex satisfying $H(E) \leq \log(n) + \mathcal{O}(1)$, and $f$ is sampled according to $p_f$ satisfying $\left\| \frac{p_f}{p_U} \right\|_\infty \leq n^c$ for some constant $c$, where $p_U$ is a uniform distribution (Compton et al., 2022).

In assumption (Aii) above, we use the term sufficiently close to refer to the existence of a sufficiently small upper bound on the distance between the two values.

Assumptions $(Ai)$ and (Aiii) are primarily technical assumptions to ensure applicability of inequalities 7 and 9 used in the proof. Assumption (Aii) on the other hand, while facilitating steps of the proof, has a conceptual implication: (Aii) implies that the difference in the randomness of the two datasets is primarily due to $X_1$ and $X_2$.

### C.2  PROOF

The proof of Theorem 2, under the assumptions above, relies on Inequality 7 from Pouryahya et al. (2017) and Lott & Villani (2009), and the results from Compton et al. (2020) and Compton et al. (2022) leading to Inequality 9. Given alternative models $X_i = g(Y_i, \tilde{E}_i)$ for $i = 1, 2$ with exogenous variables $\tilde{E}_i$, under (Aiii), Inequality 9 gives the following lower bound on the Shannon entropy of $\tilde{E}_i$,

$$H(X_i) - H(Y_i) + H(E_i) < H(\tilde{E}_i). \qquad (13)$$

Suppose $\underline{k}_i < 0 \leq \underline{k}_2$ where $\underline{k}_i$ is a lower bound on the Ricci curvature corresponding to $X_i$. Then, by Inequality 7, assuming (Ai), we have

$$\underline{s}_2 > \underline{s}_1, \qquad (14)$$

where $\underline{s}_i$ is a lower bound on the Boltzmann entropy corresponding to $X_i$. On the other hand, the Boltzmann entropy can be written as a constant scaling of the Shannon entropy. Thus, given lower bounds $\underline{h}_1$ and $\underline{h}_2$ on $H(X_1)$ and $H(X_2)$, Inequality 14 implies $\underline{h}_2 > \underline{h}_1$. Consider a constant $h \in (\underline{h}_1, \underline{h}_2)$. Since $\underline{h}_2$ is a lower bound for $H(X_2)$, it holds that $\mathbb{P}\left[H(X_2) \geq h\right] = 1 \geq \mathbb{P}\left[H(X_1) \geq h\right]$, where $\mathbb{P}(.)$ denotes the probability. Hence, under assumption (Aii), $\mathbb{P}\left[\Lambda_2 > \eta\right] = 1 \geq \mathbb{P}\left[\Lambda_1 > \eta\right]$, where $\Lambda_i := H(X_i) - H(Y_i) + H(E_i)$ and $\eta \in (\underline{h}_1 - H(Y_1) + H(E_1), \underline{h}_2 - H(Y_2) + H(E_2))$ is a constant. Using the lower bounds in 13, this implies

$$\mathbb{P}\left[H(\tilde{E}_2) > \eta\right] \geq \mathbb{P}\left[H(\tilde{E}_1) > \eta\right],$$

completing the proof of the theorem connecting causal inference with curvature.

## D  RELATED WORK

We showed the connection between curvature and causal inference, bridging for the first time the works on invariance and robustness of causal models, geometric deep learning, and deep learning for causal inference. Although we have cited related works in each section as appropriate, we now mention the most relevant works in each aspect and reference other contributions in the literature. While these works provide essential foundations and motivations for the theory in this work, none of them establishes the explicit links we developed in the paper and, in particular, the close connection between geometry/curvature, robustness, and causal inference.

**Invariance, Robustness, and Causal Inference.** Learning representations that are invariant across a set of environments is the primary goal of invariant causal prediction (ICP) (Bühlmann, 2020; Peters et al., 2016; Heinze-Deml et al., 2018; Shi et al., 2021) and invariant risk minimization (IRM) (Bühlmann, 2020; Shi et al., 2021; Arjovsky et al., 2019; Lin et al., 2022). Bühlmann (2020) formally describes how IRM can lead to a distributionally robust estimator while imposing causal identification assumptions.

**Geometric Deep Learning.** Geometric tools have been instrumental to recent advances on GNNs (Bruna et al., 2014; Bronstein et al., 2017; Gong et al., 2020; Bronstein et al., 2021). Discrete Ricci curvatures on graphs, in particular, are well-established measures with roots in Riemannian geometry (Ollivier, 2009; Sandhu et al., 2015; Samal et al., 2018), with numerous applications for GNNs (Topping et al., 2021; Southern et al., 2023). The connection between Ricci curvature and entropy is known from the optimal transport literature (Lott & Villani, 2009), based on which, (Pouryahya et al., 2017) uses Ricci curvature as a measure of system robustness. Moreover, curvature has been used by Srinivas et al. (2022) to improve robustness in neural networks. However, the literature does not establish a connection with distributional robustness, a gap that we fill with the help of results from entropic causal inference (Kocaoglu et al., 2017; Compton et al., 2020; 2022).

**Deep Learning for Causal Inference.** Deep learning methods have had success in estimating treatment effect (Shalit et al., 2017; Louizos et al., 2017), counterfactual inference (Johansson et al., 2016; Pawlowski et al., 2020), and other problems in causal inference (Luo et al., 2020; Lu et al., 2021; Perry et al., 2022; Frauen & Feuerriegel, 2022; Ke et al., 2022; Immer et al., 2023; Hägele et al., 2023). Causal effect estimation on networked data on the other hand, is known to be notoriously challenging (van der Laan, 2012; Zheleva & Arbour, 2021). Various methods have been proposed for estimating the causal effect in structured data which violate traditional identification assumptions (Jiang & Sun, 2022; Kaddour et al., 2021; Cristali & Veitch, 2022; Ma & Tresp, 2021; Guo et al., 2020; Harada & Kashima, 2021). For instance, Guo et al. (2020) uses a *Network Deconfounder* to learn a representation of hidden confounders from the data, Kaddour et al. (2021) proposes an effect decomposition, and Veitch et al. (2019) and Cristali & Veitch (2022) use the embeddings to deal with unobserved confounders and the homophily effect. Another approach, taken in Jiang & Sun (2022); Ma & Tresp (2021); Harada & Kashima (2021), is to account for the peer treatment effects in the network using GNN-based causal estimation methods, which allows the violation of SUTVA. However, the literature lacks a practical indicator of the local reliability of the estimates. We show that Ricci curvature can serve as such an indicator, and informed by this result, we propose a preprocessing using the Ricci flow to improve the causal effect estimates obtained from GNN-based methods.

# E    THE NETEST MODEL

Given a graph $G = (V, E)$ with the adjacency matrix $A$, features $X$, observed outcome $Y$, and treatments $T$, the NetEst model (Jiang & Sun, 2022) uses a summary function $Z : 2^T \rightarrow [0, 1]$ to capture the peer effect on unit $v \in V$ through $v$'s *peer exposure*, $z_v = Z(\{t_u\}_{u \in N_v})$, where $N_v$ denotes the set of immediate neighbors of the node $v$. Assuming a Markov-type property that the peer effect can be learned from the signals received from immediate neighbors, the peer exposure function is set to be the average treatment of the neighbors, i.e., $z_v = \sum_{u \in N_v} t_u / |N_v|$. The ITE, $\tau(x_v)$, for two treatments $t'$ and $t''$ is then defined as

$$\tau(x_v) := \mathbb{E}\left[Y_v | do(t_v = t', z_v = z') - Y_v | do(t_v = t'', z_v = z'') \,|\, x_v, \{x_u\}_{u \in N_v}\right], \quad (15)$$

which is identified under the assumptions described in Appendix A (Jiang & Sun, 2022).

NetEst consists of four modules: an encoder, two regularizers, and an estimator. The *encoder* module learns a representation for the nodes using a graph convolutional network, producing an embedding $s_v = \phi(x_v, \{x_u\}_{u \in N_v}) \in S$ for every unit $v \in V$. The *estimator* module is trained to estimate the observed outcome from the embeddings $\{s_v\}_{v \in V}$ by minimizing a mean squared error (MSE) loss. This MSE loss $\mathcal{L}_m$ is the *potential outcome loss*, between $m(s_v, t_v, z_v)$ and the potential outcome $Y_v | do(t_v, z_v)$, where $m : S \times \{0, 1\} \times [0, 1] \rightarrow \mathcal{Y}$ denotes the estimator, assuming a binary treatment, and $\mathcal{Y}$ is the outcome space. The $p(t|x)$ and $p(z|x, t)$ *regularizer* modules are used in an adversarial training scheme to resemble randomized treatment assignment and uniform peer exposure, respectively, minimizing two MSE losses $\mathcal{L}_t$ and $\mathcal{L}_z$ on the embeddings and treatments. Hence, NetEst is trained by first training the discriminators in the regularizer modules, minimzing their respective loss values, then updating the estimator to minimze $\mathcal{L}_m$, and in the end, updating the encoder to optimize a total loss $\mathcal{L} = \mathcal{L}_m + \alpha_t \mathcal{L}_t + \alpha_z \mathcal{L}_z$.

# F    IMPLEMENTATION PARAMETERS AND HARDWARE SPECIFICATIONS

Since the main purpose of our experiments was to inspect the joint distribution of estimation errors and evaluate the impact of our proposed data preprocessing, we followed the parameters and setup used by Jiang & Sun (2022) for all implementation and training purposes of NetEst, TARNet, CFR, and NetDeconf. The encoder of NetEst contains 1 graph convolution layer, the estimator has 3 fully-connected hidden layers of size 32, and the two regularization terms in the total training loss of the encoder both have weight 0.5. The learning rate is 0.001 for 300 epochs of full batch training using an Adam optimizer (Kingma & Ba, 2015). The meta learner baselines with GNN encoders, T-learner+GNN and X-Learner+GNN, are implemented using a graph convolutional network followed by a three-layer multilayer perceptron. All meta learners were fine tuned with grid search. The system specifications for the experiments are reported in Table 2.

Table 2: System specifications for the experiments.

| | |
|---|---|
| CPU | Intel(R) Xeon(R) CPU @ 2.20GHz |
| GPU | Nvidia V100 |
| OS | Ubuntu 22.04.2 LTS |
| Architecture | x86_64 |

# G    DATA

Validating causal inference methods and theories through experiments often requires data that contain counterfactual outcomes. To this end, the standard practice in the literature is to use semi-synthetic data, where the features are empirically observed, while the treatments and potential outcomes are simulated (Hill, 2011; Shalit et al., 2017; Jiang & Sun, 2022; Veitch et al., 2019; Guo et al., 2020; Ma et al., 2021). Jiang & Sun (2022) use the BlogCatalog (BC) and Flickr datasets (Guo et al., 2020; Ma et al., 2021) to evaluate the performance of NetEst. In addition to these two datasets, we supplement our experiments with additional network datasets used in the geometric deep learning and GNN literature: Cornell, Texas, and Wisconsin networks from the WebKB dataset

[1]; Chameleon network from the Wikipedia networks dataset (Rozemberczki et al., 2021); Cora and CiteSeer networks (Yang et al., 2016); and Actor network (Pei et al., 2019). Table 3 includes descriptive statistics on these networks. Note that we only use the largest connected component in each network. Following Jiang & Sun (2022), we split each network data into training, validation and test sets using METIS (Karypis & Kumar, 1998). The treatments and potential outcomes for all network data are synthesized following the formulation in Jiang & Sun (2022).

Table 3: Descriptive statistics for the networks used in our experiments.

|          | BC     | Flickr | Cornell | Texas | Wisconsin | Chameleon | Cora  | CiteSeer | Actor |
|----------|--------|--------|---------|-------|-----------|-----------|-------|----------|-------|
| Nodes    | 5196   | 7600   | 183     | 183   | 251       | 2277      | 2708  | 3327     | 7600  |
| Edges    | 171743 | 30019  | 298     | 325   | 515       | 36101     | 10556 | 9104     | 30019 |
| Features | 8189   | 932    | 1703    | 1703  | 1703      | 2325      | 1433  | 3703     | 932   |

## H    FURTHER EXPERIMENTS

### H.1    RICCI CURVATURE AND TREATMENT EFFECT ESTIMATION ERROR

In this section we include additional plots showing the joint distributions of the ITE estimation error for each node $v \in V$, $\varepsilon_{ITE}(v)$, and the Ricci curvature in the neighborhood of the node, for both Forman and Ollivier Ricci curvatures (this one repeated from Figure 2 for completeness and ease of visualization/comparison). The distributions for the nine networks are shown in Figure 3, with two plots (one per curvature) for each dataset. All ITE estimations in this figure have been obtained using NetEst (Jiang & Sun, 2022). These distributions and the regression lines marked on the plots further confirm our theoretical results, which imply that highly negative Ricci curvature makes causal effect estimation more challenging.

### H.2    ITE ERROR DISTRIBUTION

In order to obtain a better understanding of how the Ricci flow adjustment impacts ITE estimation for each unit, we compare the empirical cumulative distribution functions (CDFs) of $\varepsilon_{ITE}$ obtained from f-NetEst and NetEst, in the two datasets used by Jiang & Sun (2022), as well as seven other networks described in Appendix G. As shown in Figure 4, the empirical CDF from f-NetEst is uniformly above that from NetEst for low $\varepsilon_{ITE}$ values, which further confirms that flattening the edges leads to a larger proportion of units with low ITE estimation error.

---

[1]http://www.cs.cmu.edu/afs/cs.cmu.edu/project/theo-11/www/wwkb/

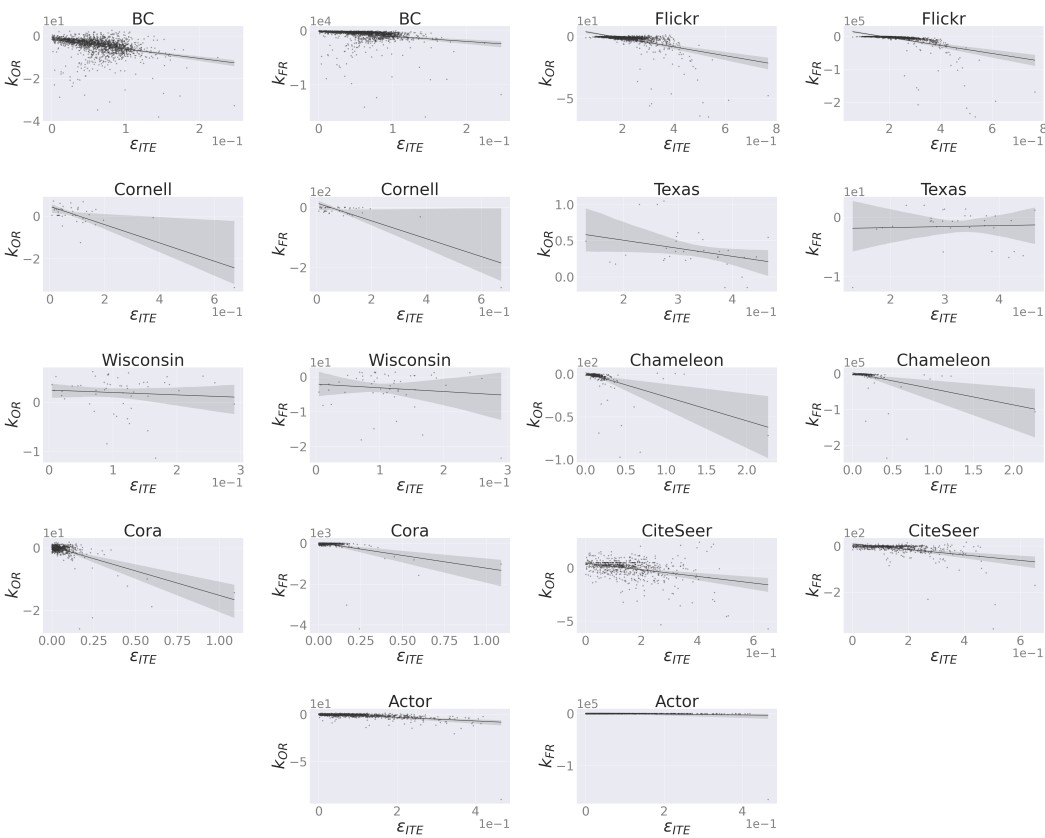

Figure 3: Joint distributions of the sum of Forman and Ollivier-Ricci curvatures in the neighborhood of each node and the estimation error of ITE for that node. The distributions for the nine networks are shown, with two plots (one per curvature) for each dataset. The regression lines with the corresponding 95% confidence intervals are marked on the plots.

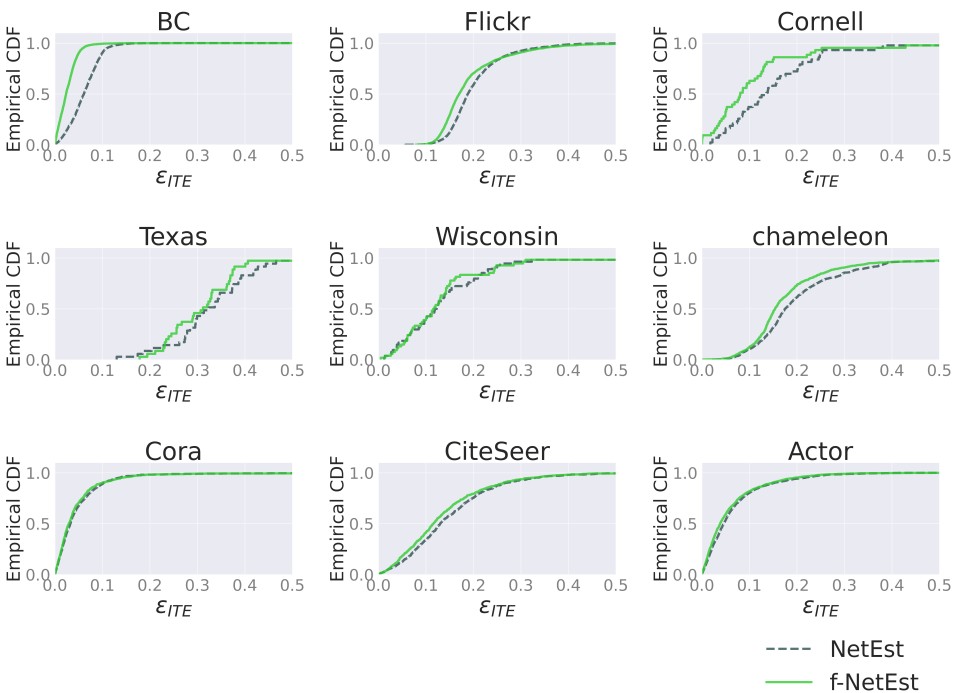

Figure 4: Empirical CDF of the ITE error, $\varepsilon_{ITE}$, obtained from NetEst (black) and f-NetEst (green).

