# OpenReview forum: "Ricci Curvature, Robustness, and Causal Inference on Networked Data"
_ICLR.cc/2024/Conference — ICLR 2024 Conference Withdrawn Submission_

### Official Review · Reviewer_fMiq · 2023-10-30

**Soundness:** 2 fair
**Presentation:** 2 fair
**Contribution:** 3 good
**Rating:** 5
**Confidence:** 4

**Summary:**

The authors explore networked data via graph-neutral networks. Using graph Ricci curvature, they predicted the reliability of causal effect estimations on networked data. Their empirical findings confirmed that positive curvature regions yield more accurate results.

**Strengths:**

The main review will take place in this section owing to the flow in which the review was conducted.

### Abstract

- The first sentence is interesting and whilst I know this is the abstract, it would be helpful to know _why_ the author(s) think it is critical challenge to understand the causal effect in networked data - who cares and why?


### Introduction

- What do you mean by "the endogeneity"  - this is a fairly non-standard way to use that way (at least to this reviewer, hence please do tell your usage of the term)
- "identifying causal effects is particularly challenging on a network of units with non-trivial dependencies" - why is this? What gives rise to the difficulty?
- A graphic may be helpful at the end of page one to illustrate the concepts that you discuss (e.g. graph curvature) for readers less familiar with this topic.


### CAUSALITY, INVARIANCE, AND ROBUSTNESS

- What is meant by "unit" in the first sentence of this paragraph?
- Do you assume the presence of unobserved confounders? If so, please spell that out.
- Why is your outcome denoted $\mathbf{Y}$? That gives the impression that you are dealing with a multi-objective (more than one outcome) setting - are you? Perhaps best to spell out the domain of the features and the outcome(s) either way to avoid any confusion.
- I find definition 2 and the paragraph following, very vague and ambiguous. It is not clear what you are describing. What is $\mathbf{C}$? The way the bottom part of page 3 is phrased, make it difficult for a reader to extract meaning from a language that is rather too formal (without providing all the background, since that, I assume, lives in (Buhlmann, 2020)).
- Whilst I appreciate that (Rothenhausler et al., 2021) uses the term "anchor" for a very common graph concept, it may be better to stick with the common tongue when it comes to graph formalism, where $A$ in figure 1 is either a leaf node or a root node depending on the perspective of the modeller. These terms predate the former's by a few decades and so may make this part of the section of the paper a bit more accessible by using the more common nomenclature.
- You should adopt the standard graphical norms when it comes to causal inference. If H is a hidden confounder then it is standard to dash the edges (H,X) and (H,Y). The way you have drawn figure 1 makes it appear as if H is observed.
- H is hidden (latent or unobserved) yet you are including it in equation 2? Please explain how that works.
- You use 'source' and 'anchor' to refer to variable $A$ but never the standard 'root/leaf'? Consider picking one of these four terms.

### CURVATURE, ROBUSTNESS, AND ENTROPY

- The start of section 3 could do with an illustration of the concepts involved with Ricci curvature (e.g. figure four from the top: https://www.researchgate.net/publication/334371953_Community_Detection_on_Networks_with_Ricci_Flow/figures?lo=1)
- Use $S(\cdot)$ and not $S(.)$ for the Boltzmann entropy.

### Entropic causal inference

- Move the review of entropic causal inference to the preliminaries - the introduction of it the way you have done it now, breaks the flow of the paper.
- This does make any sense: $Y = f(X,E)$ - what is $f$? What is $X$ the same as before? What is $E$ noise terms? What is the causal modelling framework you are using? Is $f$ an SEM? You seem to suggest that $E$ are exogenous variables which d-separeted (?) I assume that's what your symbol means since you haven't defined it, from the outcome variable $Y$ but then you introduce something you call 'alternative exogenous variables' - what's that?
- Equation 9 presently does not point toward anything since it is not clear how the connection is made between it and what you say it is pointing to, which is not clear since section four is lacking a lot of detail for this to be a convincing argument.
- What I find curious further about this section if that if equation 9 is a powerful result you build on, it would be helpful if you placed it in a lemme/theorem/proposition to indicate its importance to the paper.

### Curvature and causal inference

- Paragraph 2 in this section could use a figure for your stated causal relationship.
- This may be a very silly question but you say: "we expect a higher entropy of the exogenous variables when the curvature is larger" - how can you expect anything at all about the exogenous variables since you cannot measure them? They are latent and so unknown to the modeller. Hence, what use is theorem 2 (being the devil's advocate here)
- Suggestion: place this within a remark: "This ultimately suggests that more positive Ricci curvatures are expected to correspond to lower errors in estimating the causal effect"

### RELATED WORK

- What are you evaluation metrics actually measuring w.r.t. your contribution in this paper? What is the relation to Ricci curvature w.r.t. these metrics?
- To confirm, none of your networks have unobserved confounders?

**Weaknesses:**

See the Strengths section for relevant comments.

**Questions:**

See the Strengths section for relevant comments.

---

> ### Author Response · Authors · 2023-11-18
>
> We thank the reviewer for their feedback, which we have carefully considered. Recognizing that certain aspects of our paper may not have been fully understood, we hope that the following clarifications, albeit brief, will shed some light on these points. We remain committed to enhancing the clarity and presentation of our paper and we will do that in resubmission.
>
> We appreciate the comments and questions posed by the reviewer and we will use these to clarify both the background concepts and the contribution of the paper. Here we briefly clarify some of the points, concepts, and terminology in our paper, following this reviewer’s comments and questions. The ‘endogeneity’ in the network structure refers to the situation where network structure interferes with the causal model, i.e. the connections between units are endogenous to the model between treatment and outcome of the units, and hence interferes with the causal mechanism. Regarding unobserved confounders, the f-NetEst method uses the NetEst model for estimating treatment effect, and hence, follows the same identification strategy described in our Appendix A. This includes a variation of ignorability, assuming away any unobserved confounders. Regarding the section on entropic causal inference, Equation 9 is used in proving Theorem 2. We understand that the discussion in this section could benefit from clarification on the SEMs and the variables described, e.g., the brief clarification we provided in our response to Reviewer UoX4. We will improve this discussion. We further acknowledge that elaborating on the discussion following Theorem 2 could help clarify its implications for the reader, which will also resolve the question this reviewer has raised about its conclusion relying on the entropy of exogenous variables.
>
> We hope these explanations address some of the questions raised by the reviewer. We again thank the reviewer for their questions and comments. It is unfortunate that as the reviewers expressed, they did not fully understand the paper. We take responsibility for that and will use their comments for improving the presentation.

---

### Official Review · Reviewer_poe6 · 2023-10-31

**Soundness:** 2 fair
**Presentation:** 1 poor
**Contribution:** 1 poor
**Rating:** 3
**Confidence:** 5

**Summary:**

This paper claims to address the challenge of understanding causal effects in networked data using Graph Neural Networks (GNNs). They claim to utilize the link between graph curvature and causal inference, finding that negative curvatures complicate identifying causal effects to predict the reliability of causal effect estimations, showing that positive curvature regions lead to more accurate results.

**Strengths:**

The exploration of curvature is intriguing.
Causal inference on network data is an interesting problem to investigate.

**Weaknesses:**

- The contribution of this paper is relatively moderate, as it amalgamates ideas from other sources, resulting in a contribution that is at best modest.
- The paper is poorly written.
- This paper seems to essentially be using causal discovery to improve causal inference. This is not properly explained.
- Sounds like this paper violates the SUTVA assumption in casual inference on networks.

**Questions:**

- This paper seems to essentially be using causal discovery to improve causal inference.  Could the author please elaborate on that?
- How do you justify violating the SUTVA assumption in casual inference on networks?
- Implementing Ricci flow adjustments involves complex computations, especially on large-scale networks, which can be computationally intensive and time-consuming. This complexity may limit its applicability to real-time or resource-constrained scenarios. How do the authors justify the complexity and computational costs?
- How interpretable are the results? What is behind the improved estimations (beyond statistical metrics)?
- Ricci flow adjustments' effectiveness varies based on the unique structure of real-world networks, which are diverse and dynamic. Generalizing this method across different scenarios is challenging due to the complex nature of network connections. Moreover, the method's reliance on parameters like edge weights and curvature values makes it highly sensitive to even minor fluctuations. Selecting appropriate parameter values demands careful tuning and a deep understanding of the network, posing significant challenges in achieving optimal performance. Could the authors elaborate on these issues?

---

> ### Author Response · Authors · 2023-11-18
>
> We thank the reviewer for their feedback, which we have carefully considered. Recognizing that certain aspects of our paper may not have been fully understood, we hope that the following clarifications, albeit brief, will shed some light on these points. We remain committed to enhancing the clarity and presentation of our paper and we will do that in future submissions.
>
> We appreciate that the reviewer has pointed to our use of GNN in empirical validation of the connection we establish between geometric deep learning and causal inference. This paper establishes, for the first time, a theoretical connection between graph geometry and causal inference on networks and proposes a method for improving causal effect estimates based on this theoretical result. The theoretical findings are strongly supported by our empirical results, which are obtained through integrating our proposed method with previously published GNN-based models for treatment effect estimation.
>
> We appreciate the comments and questions posed by the reviewer and we will use these to clarify both the background concepts and the contribution of the paper. To point the reviewer to answers regarding their question about causal discovery, we refer this reviewer to our response to Reviewer UoX4. About the violation of SUTVA, as we have mentioned in the manuscript (last sentence of Section 2.1 and first paragraph of Section 7.1, in addition to Appendix A), the violation of SUTVA in network settings is in fact often a motivating starting point of network causal inference methods, such as the NetEst model used in our experiments. In other words, the identification strategy of these methods (as described in our Appendix A) does not rely on SUTVA. Regarding the improvement in estimation of causal effects, this is mainly due to the close connection between curvature and robustness, which we briefly pointed to in the paper and will further clarify.
>
>
> We hope these explanations address some of the questions raised by the reviewer. We again thank the reviewer for their questions and comments. It is unfortunate that as the reviewers expressed, they did not fully understand the paper. We take responsibility for that and will use their comments for improving the presentation.

---

### Official Review · Reviewer_UoX4 · 2023-11-06

**Soundness:** 2 fair
**Presentation:** 2 fair
**Contribution:** 3 good
**Rating:** 3
**Confidence:** 3

**Summary:**

This manuscript bridges the Ollivier Ricci curvature with causal inference and shows that positive curvatures 'help' with the causal inference while negative ones do not. The connection is built on a causal model that regards the problem as a worst-case risk minimization. The main spot of the theoretical analysis is to show that if the ricci curvature is negative, then higher entropy is needed to fit the causal model. Numerical results align with the theoretical analysis. Further experiments show that integrating ricci flow with GNNs can improve the performance.

**Strengths:**

1. The paper discusses the background and related work quite carefully, and is in general well-written.
2. I like this discovery. Building connections between geometry of the network and the causal inference should shed light on more future work, in addition to more understanding of this topic, especially given the fast development.
3. I acknowledge the theoretical analysis as the main contribution, it is good that the experimental results corroborate with the theoretical analysis.

**Weaknesses:**

1. It seems that the background is discussed too much (until page 5) and the main entree is too short (only one page), which possibly indicates that this work relies a lot on previous work. Further, the large body of discussion triggers confusion, I would expect a clear logical argument, for example, the causal model has some property, leads to robustness, further goes to entropy, and lands in Ricci curvature. The above may be wrong, but readers need the correct version instead of a stack of notions.

2. As for the proposed method integrating ricci flow and GNN, I could not find anything else except one paragraph in section 5.1. So if I am not misunderstanding, the proposed algorithm is: First run Ricci flow and attain the edge weights, then use existing frameworks of GNN-based causal inference. This is OK but the contribution is quite limited. Have the authors think about how to further take advantage of this property and design new GNN frameworks? Such as [1] and many others. Further, it would be good to at least present the pseudo code/descriptions by steps of the proposed method.

[1] Curvature Graph Networks.

**Questions:**

1. I unfortunately do not understand the formalization part very well. Could you explain what are the roles of system robustness and distributional robustness in the causal model? (Especially the risk minimization). In section 3.2 you said "we utilize the correlation between Boltzmann entropy and Ricci curvature to formally establish this anticipated connection" but later in section 4, Shannon entropy is discussed and Theorem 2 also uses Shannon entropy. This is confusing, could you please make it clear?

2. I am fine with the most proof of Theorem 3.2 but not the second assumption. If I understand correctly, $X$ is the feature and $Y$ is the label, then the difference of two datasets may not just lie on the features. I wonder if the assumption that $H(Y|X;E)$ remains similar is OK in a causality task, but nothing more.

3. What does it mean by a "wrong causal model"?

4. This is not a major issue, but have you experienced computational issues when the networks are large? The computation of ricci flow takes $\tilde{O}(mn^3)$ if I remember correctly, which can be larger than the time for GNN. Then if we think about applications, this is a hurdle.

---

> ### Author Response · Authors · 2023-11-18
>
> We thank the reviewer for their feedback, which we have carefully considered. Recognizing that certain aspects of our paper may not have been fully understood, we hope that the following clarifications, albeit brief, will shed some light on these points. We remain committed to enhancing the clarity and presentation of our paper and we will do that when we re-submit the work.
>
> We appreciate the reviewer’s acknowledgement of the novel theoretical connection established by this paper between graph geometry and causal inference on networks. This is a foundational contribution, which, as the reviewer has pointed out, opens the door to harnessing the rich toolkit and insights of geometric deep learning for causal inference and causal representation learning. As such, the theoretical build up to the main contribution was intended to prepare the reader, who is not necessarily familiar with causal inference or geometry of networks. We merged multiple ML fields and very few have background in all of them. The significance of the theoretical connection is evident through the method proposed based upon it, for adjusting the input of GNN, and well-supported by empirical results.
>
> We appreciate the comments and questions posed by the reviewer and we will use these to clarify both the background concepts and the contribution of the paper. Regarding the second assumption for the theorem, we would like to briefly mention that the main implication of the theorem is how the error in causal effect estimates relates to the structure of the relationship between units, and in particular, curvature. While the assumption could very well be violated by a pair of $Y_1$ and $Y_2$ with substantially different entropies, it is reasonable to imagine that this difference also impacts the estimation of causal effects. As a result, the impact of the geometry on this estimation is not best reflected in such a scenario. Regarding what the paper means by the “wrong causal model”, suppose $X$ causes $Y$ and the SCM $Y = f(X, E)$ captures the causal effect of $X$ on $Y$. This means if an intervention sets $X=x$, the value of potential outcome $Y$ changes as predicted by $f$. However, if the observations fit an alternative model $X = g(Y, \tilde{E})$ when $Y$ is not in fact a cause for $X$, this “wrong model” does not yield correct values upon an intervention $Y=y$.
>
> We hope these explanations address some of the questions raised by the reviewer. We again thank the reviewer for their questions and comments. It is unfortunate that, as the reviewers expressed, they did not fully understand the paper. We take responsibility for that and will use their comments for improving the presentation.